# Autocrine Factors Produced by Mesenchymal Stem Cells in Response to Cell–Cell Contact Inhibition Have Anti-Tumor Properties

**DOI:** 10.3390/cells12172150

**Published:** 2023-08-26

**Authors:** Jerry P. Chen, Rong Li, Jean X. Jiang, Xiao-Dong Chen

**Affiliations:** 1Department of Molecular Medicine, University of Texas Health Science Center at San Antonio, San Antonio, TX 78229, USA; jerrypeacechen@gmail.com; 2Department of Biochemistry & Molecular Medicine, School of Medicine & Health Sciences, The George Washington University, Washington, DC 20037, USA; rli69@email.gwu.edu; 3Department of Biochemistry and Structural Biology, University of Texas Health Science Center at San Antonio, San Antonio, TX 78229, USA; jiangj@uthscsa.edu; 4Department of Comprehensive Dentistry, University of Texas Health Science Center at San Antonio, San Antonio, TX 78229, USA; 5Research Service, South Texas Veterans Health Care System, Audie Murphy VA Medical Center, San Antonio, TX 78229, USA

**Keywords:** Mesenchymal stem cells, anti-tumorigenesis, inhibitory effects of cell–cell contact inhibition on tumor cell growth

## Abstract

Recently, mesenchymal stem cell (MSC) therapies have been questioned as MSCs are capable of both promoting and inhibiting tumorigenesis. Both MSCs and tumor cells replicate to increase their population size; however, MSCs, but not tumor cells, stop dividing when they reach confluence due to cell–cell contact inhibition and then differentiate. We hypothesized that contact inhibition results in the production of effector molecules by confluent MSCs and these effectors are capable of suppressing tumor cell growth. To test this hypothesis, we co-cultured breast cancer cells (MDA-MB-231) with either confluent or sub-confluent bone-marrow-derived MSCs (BM-MSCs); in addition, we treated various tumor cells with conditioned media (CM) obtained from either confluent or sub-confluent BM-MSCs. The results showed that the growth of tumor cells co-cultured with confluent BM-MSCs or treated with CM obtained from confluent BM-MSCs was inhibited, and this effect was significantly stronger than that seen with tumor cells co-cultured with sub-confluent BM-MSCs or CM obtained from sub-confluent BM-MSCs. Subcutaneous tumor formation was completely prevented by the inoculation of tumor cells mixed with CM. In the future, soluble anti-tumor effectors, produced by confluent MSCs, may be used as cell-free therapeutics; this approach provides a solution to current concerns associated with cell-based therapies.

## 1. Introduction

Mesenchymal stem cells (MSCs) serve as a “cell reservoir”, continuously replacing old cells to maintain/repair tissues damaged by trauma, disease, or aging, and as a “drug store”, secreting trophic factors which are anti-inflammatory and capable of treating autoimmune diseases. Over the years, the use of MSCs in cell-based therapies has faced scrutiny due to a number of conflicting studies that have shown that MSCs are capable of both promoting and inhibiting tumorigenesis [1,2,3]. As a result, cogent arguments have been proposed which advise against the use of MSCs in treating age-related diseases or in cases of recovering cancer patients receiving high-dose chemo- or radiotherapy. 

During aging, both stem cell number and quality decline [4,5,6,7]. Over the last 3 decades, MSCs have been extensively studied and shown to play a critical role in maintaining tissue homeostasis and regeneration [8,9,10]. Since many malignancies are also considered to be age-related diseases, it is theoretically possible to inhibit tumorigenesis if a sufficient number and type of high-quality stem cells could be provided. However, the biomedical literature is replete with conflicting reports showing that in some cases MSCs promote tumorigenesis, while in other cases, they inhibit tumorigenesis. This issue must be resolved before the use of MSCs as potential therapeutics for age-related diseases can be further explored. 

A thorough review of the literature provides a logical explanation for the inconsistent tumorigenic/anti-tumorigenic behavior of MSCs [1,2,3]. For example, MSCs have been reported to *promote* tumorigenesis by secreting interleukin-8 (IL-8), IL-10, and indoleamine 2,3-dioxygenase (IDO) to suppress the immune system and by secreting vascular endothelial cell growth factor (VEGF), transforming growth factor-beta (TGF-β), and IL-6 to stimulate tumor vascularization [11]. In contrast, MSCs have been shown to have anti-tumorigenic effects by releasing tumor necrosis factor-related apoptosis-inducing ligand (TRAIL) which selectively induces apoptosis in various cancers by inhibiting extracellular signal-regulated kinase-1/2 (ERK1/2) and protein kinase B (PKB)(also known as AKT) action and Dickkopf-related protein-1 (DKK-1) that block tumor growth by suppressing Wnt signaling [12,13,14]. Since MSCs are known to be exquisitely responsive to their local microenvironment, it is plausible that the variable results may be attributed to differences in the experimental conditions used to conduct the studies. Typically, two primary in vitro approaches have been used to study the effects of MSCs on tumor cell growth. In the first approach, MSCs and tumor cells are co-cultured either in direct cell–cell contact or separated by a semi-permeable membrane allowing exposure to diffusible soluble factors. In the second approach, tumor cells are exposed to MSC-conditioned media (CM) during culture. Conflicting results have also been found when in vivo approaches are used [2,3]. Typical in vivo experiments have employed two different strategies for administering the cells. In the first approach, cancer cells and MSCs are mixed together and then injected into a single site (referred to as “co-injection”), while in the second approach, cancer cells and MSCs are injected into two separate and distant sites (referred to as “distant injection”). The results obtained using these approaches are further complicated by several factors, including: (1) variations in donor age and anatomical site where the MSCs were obtained (e.g., young, middle-aged, and old donors; umbilical cord-, adipose-, or bone-marrow-derived MSCs), (2) how the MSCs were prepared (e.g., harvesting procedures, culture conditions, number of passages in sub-culture, proliferation versus differentiation, etc.), (3) ratio of MSCs to tumor cells seeded in vitro or injected in vivo, (4) features of the animal model being employed (e.g., species, age, and gender), (5) how the cells are administered in vivo (timing and route), and many other considerations. Unfortunately, in almost all of these studies, the effect of MSC maturation state on the tumor cells has received less attention. Since it is well known that the properties of MSCs vary with their cell maturation state, the influence of MSCs on tumorigenesis would be maturation-dependent. 

Both MSCs and tumor cells undergo replication or self-renewal to increase their population size. However, once they reach confluence MSCs, but not tumor cells, stop dividing and initiate differentiation due to cell–cell contact inhibition. In the present study, we hypothesize that contact inhibition results from the production of effectors by confluent MSCs and these effectors are also capable of suppressing tumor cell growth. The current report provides evidence, for the first time, suggesting that confluent MSCs produce inhibitory factor(s) that strongly suppress the proliferation of a variety of tumor cell lines in vitro as well as tumor growth in vivo. 

## 2. Materials and Methods

### 2.1. Human Tumor Cell Lines and Skin Fibroblasts

The following human tumor cell lines were purchased from the American Type Culture Collection (ATCC, Manassas, VA, USA) and expanded as described by the supplier before use in the experiments: MDA-MB-231 (a human breast cancer cell line), MG63 (a human osteosarcoma cell line), CaLu-6 (a human lung adenocarcinoma cell line), HT-29 (a human colorectal adenocarcinoma cell line), and PC3 (a human prostatic adenocarcinoma cell line). Human neonatal foreskin fibroblasts were also purchased from ATCC and expanded as described by the supplier before use in the experiments.

### 2.2. Preparation of Bone Marrow-Derived Mesenchymal Stem Cells (BM-MSCs)

Freshly isolated human bone marrow (BM) mononuclear cells from healthy young donors (≤ 25 years old) were purchased from Lonza Group Ltd. (Walkersville, MD, USA), and BM-MSCs were prepared as previously described [15,16]. Briefly, fresh BM mononuclear cells, containing BM-MSCs, were seeded onto standard, tissue culture plastic (TCP) vessels (Millipore-Sigma, St. Louis, MO, USA) at 5 × 10^5^ cells/cm^2^ and cultured in growth media consisting of α-MEM (Life Technologies, Grand Island, NY) containing glutamine (2 mM), penicillin/streptomycin, and 15% fetal bovine serum (FBS) (Atlanta Biologicals, Lawrenceville, GA). One-half (i.e., 50%) changes of growth media were performed every 3–4 days until BM-MSCs appeared and formed colonies (70–80% confluence, ~2 weeks). Non-adherent cells were removed by washing with phosphate buffered saline (PBS) and adherent cells (i.e., passage 1; P1 cells) collected by trypsinization. The BM-MSC phenotype was confirmed by expression (>90%) of CD73, CD90, and CD105 and negative selection for CD34, CD45, and HLA-DR [15]. Cells were either used immediately in the experiments or frozen in liquid nitrogen (LN2) for later use.

### 2.3. Preparation of Human Umbilical Cord Mesenchymal Stem Cells (UC-MSCs) 

UC-MSCs were purchased from StemBioSys, Inc. (San Antonio, TX, USA) and expanded by culturing on TCP in growth media. Cells (P1 through P5) were used in the indicated experiments. 

### 2.4. Co-Culture Experiments 

BM-MSCs (P2-P4) were seeded into 6-well TCP plates at 4 × 10^3^ cells/cm^2^ and cultured in growth media for 4 (sub-confluent) or 7 (confluent) days. Incubation of the sub-confluent or confluent MSCs was continued for an additional 48 h (with changing to DMEM containing 10% FBS); subsequently, co-cultures were initiated by seeding MDA-MB-231 cells (3 × 10^3^ cells/cm^2^) directly onto the sub-confluent (day 6 cultures) or confluent (day 9 cultures) MSCs (“Direct”) or onto a Transwell membrane (Sterlitech, Auburn, WA, USA) and then placed in a well containing sub-confluent or confluent MSCs (“Insert”) and cultured for 5 days. At harvest, the total number of cells per well was determined; MDA-MB-231 cells were identified by GFP expression using flow cytometry. 

### 2.5. Preparation of BM-MSC Conditional Media (CM) for Treating Human Tumor Cells

BM-MSCs (P2-P4) were seeded into 6-well TCP plates at 4 × 10^3^ cells/cm^2^, cultured in growth media for 4 (sub-confluent) or 7 (confluent) days, and then transferred to serum-free αMEM media. After 48 h, the supernatant (or CM) was collected from either sub-confluent or confluent BM-MSCs in culture, centrifuged at 400× *g* for 20 min, and then filtered through a 0.45 μm microporous membrane. 

The ability of CM, collected from pre- and post-confluent BM-MSCs, to inhibit the proliferation of MG63 osteosarcoma cells or neonatal foreskin fibroblasts was determined. Briefly, cells were plated into 96-well plates at 1–3 × 10^3^ cells/cm^2^. After 48 h, the media were changed to media containing 0%, 1%, 10%, and 30% CM and cell number determined after 5 days of culture using the MTT assay (Millipore-Sigma, Burlington, MA). 

In addition, other tumor cell lines, including MDA-MB-231, CaLu-6, HT-29, and PC3 cells, were cultured separately in the presence of 10% CM. After 6 days in culture, cell number was determined. 

### 2.6. Western Blot Analysis

MDA-MB-231 cells (3 × 10^3^ cells/cm^2^) were seeded directly onto confluent BM-MSCs or a Transwell membrane and placed into a well containing confluent MSCs (“Insert”) and cultured. After 5 days, MDA-MB-231 cells were identified by flow cytometry based on GFP expression and then lysed for Western blot analysis using a phospho-specific (T180/Y182)-anti-p38 antibody (Cell Signaling), as previously described [17]. 

### 2.7. Tumor Formation Assay: Subcutaneous Inoculation

CaLu-6 lung cancer cells (1 × 10^6^) were suspended in 100μL of 10% CM or PBS and then subcutaneously injected into female immunocompetent BALB/c mice (Charles River, Wilmington MA, USA) on the right or left ventral side, respectively. The mice were humanely euthanized 3 months after injection of the cells. All animal procedures were approved by the UTHSCSA Institutional Animal Care and Use Committee. 

### 2.8. Statistical Analysis 

All data are presented as the mean ± standard deviation with an n of 3 to 6 depending on the experiment. Statistical differences were identified with use of Student’s *t* test or one-way ANOVA with significance set at *p* < 0.05. All results were reproduced in at least 3 independent experiments. 

## 3. Results

### 3.1. Confluent BM-MSCs, Maintained on Tissue Culture Plastic (TCP), Dramatically Inhibit the Growth of MDA-MB-231 Breast Cancer Cells

To compare the ability of pre- vs. post-confluent BM-MSCs to inhibit the growth of breast cancer cells, MDA-MB-231 cells prelabeled with green fluorescent protein (GFP), were seeded directly onto either sub-confluent (~70%) or fully confluent BM-MSCs grown in TCP wells (“Direct”) or onto a Transwell membrane, which was then inserted into a TCP well containing sub-confluent or fully confluent BM-MSCs (“Insert”) and then cultured for 5 days (Figure 1). The results showed that compared to culture on TCP alone, MDA-MB-231 cell density (proliferation) was dramatically decreased >95% by Direct or Insert co-culture on confluent BM-MSCs (Figure 1A,C). Although the density of MDA-MB-231 cells incubated under Direct or Insert co-culture on sub-confluent BM-MSCs was also significantly inhibited, the decrease in proliferation was attenuated by 70–80% (Figure 1A,C). 

To assess the specificity of the tumor cell response, we replicated the above experiments and co-cultured MDA-MB-231 cells with confluent human umbilical-cord-derived MSCs (UC-MSCs) or human neonatal foreskin fibroblasts. The data shown in Figure 2A,B are remarkably similar to the results obtained previously with the BM-MSCs (Figure 1). In both cases, confluent MSCs (i.e., BM- and UC- MSCs) co-cultured with MDA-MB-231 cells, either directly or indirectly using the Transwell membrane (Insert), almost completely inhibited proliferation of the tumor cells. In contrast, co-culture with confluent neonatal foreskin fibroblasts had no inhibitory effect on the tumor cells (Figure 2B).

Interestingly, MDA-MB-231 cells co-cultured with confluent BM-MSCs up-regulated phosphorylated p38 (Phos-p38) MAPK (Figure 2C), suggesting that the observed reduction in cancer cell proliferation might be associated with the up-regulated p38 phosphorylation and contact inhibition [17]. 

### 3.2. Conditioned Media Produced by Confluent BM-MSCs in Culture Inhibited the Proliferation of a Variety of Tumor Cells 

Since our data demonstrated that confluent MSCs were capable of inhibiting the growth of MDA-MB-231 cells in co-culture by both direct and indirect cell–cell contact, we hypothesized that when stem cells reach confluence and undergo contact inhibition (cease proliferation) the resulting secretion of autocrine/paracrine factors are also capable of suppressing tumor cell growth. To test this hypothesis, we collected conditioned media (CM) from confluent BM-MSCs and then treated MG63 osteosarcoma cells and neonatal foreskin fibroblasts with different dilutions (1, 10, and 30%) of the CM. Figure 3A,B show that MG63 cell growth was inhibited even at 1% CM, which resulted in an approximate 60% reduction in cell division and induced apoptosis. In contrast, the BM-MSC CM had no inhibitory effect on foreskin fibroblasts at any of the tested concentrations (Figure 3A). 

To determine if BM-MSC CM had similar inhibitory effects on other tumor cells, we selected various tumor cell lines for testing: CaLu-6 cells (a human lung adenocarcinoma cell line), HT-29 cells (a human colorectal adenocarcinoma cell line), and PC3 cells (a human prostatic adenocarcinoma cell line). MDA-MB-231 cells (a human breast cancer cell line) were used as a positive control. CM (10%) from confluent or sub-confluent cultures of BM-MSCs were added to the tumor cell cultures and incubation continued for 6 days (Figure 3C). The results show that CM inhibited cell growth in all tumor cell lines tested and CM from confluent BM-MSCs elicited a stronger inhibition than CM from sub-confluent cultures. These results are consistent with our previous results where we showed that the growth of MDA-MB-231 cells was inhibited by co-culture with confluent or sub-confluent BM-MSCs (Figure 1C). 

### 3.3. Conditioned Media Obtained from Confluent BM-MSCs in Culture Inhibited Tumor Growth In Vivo

Next, we combined 1 × 10^6^ lung cancer cells (CaLu-6) with 10% CM derived from confluent cultures of BM-MSCs or PBS (untreated controls), which were injected subcutaneously into right or left ventral sites, respectively, of an immunocompromised mouse (Figure 4). Six weeks after injection, the sites where the CaLu-6 cells were mixed with 10% CM showed almost no tumor formation in repeat experiments with 9 individual mice. In contrast, all mice injected with CaLu-6 cells mixed with PBS consistently formed tumors of various sizes.

## 4. Discussion

Typically, MSCs undergo three different stages of maturation. For example, BM-MSCs go through phases of proliferation, differentiation (extracellular matrix synthesis), and mineralization [18], and each phase is characterized by distinct cell phenotypes (e.g., self-renewal versus differentiation capacity, profile of trophic factors, etc.). The present study is built upon our prior experience with MSCs, the importance of stem cell maturation state on MSC behavior, and our review of the literature revealing a paucity of studies examining the role of MSCs and their maturation state on tumor cell growth. A previous study by Karnoub et al. reported that pre-confluent BM-MSCs co-cultured with breast cancer cells promoted tumor cell proliferation [2]. However, our data demonstrated that BM-MSCs strongly suppressed breast cancer cell proliferation when co-cultured on a confluent monolayer of the stem cells or separated by a Transwell membrane (Insert), allowing the exchange of soluble factors between the cells (Figure 1). In our studies, we specifically cultured BM-MSCs beyond confluence to cease proliferation and commence differentiation. This cessation of proliferation would have resulted in the production of a new panel of effector(s)/trophic factor(s). In contrast, the inhibitory effect of sub-confluent BM-MSCs on MDA-MB-231 cell growth was significantly attenuated (Figure 1A). Subsequently, we extended the experimental approach to include UC-MSCs and foreskin fibroblasts. Indeed, both BM- and UC-MSCs, not foreskin fibroblasts, displayed the same inhibitory effects on MDA-MB-231 cell growth in both “direct” and “indirect” (Insert) co-cultures (Figure 2B). Furthermore, our data showed that p38 mitogen-activated protein kinase (MAPK) activity was up-regulated in cancer cells co-cultured directly or indirectly (Insert) with confluent BM-MSCs (Figure 2C). This is an interesting observation since p38 MAPK has been associated with cell–cell contact inhibition, which triggers the transition from proliferation to differentiation, and may act as a suppressor of tumorigenesis [13], mediated by inhibitory factor(s) secreted by the confluent BM-MSCs.

Due to the evidence we obtained suggesting that the suppression of MDA-MB-231 cell growth was mainly caused by trophic factor(s) secreted by BM- or UC-MSCs, we treated a variety of tumor cell lines with conditioned media (CM) collected from confluent versus sub-confluent BM-MSCs. Consistent with our co-culture results, CM from confluent BM-MSCs was more effective in reducing tumor cell number (i.e., tumor cells per cm2) than CM from sub-confluent BM-MSCs. Moreover, the inhibitory effects of BM-MSC CM were found across a broad range of tumor cell types, including osteosarcoma (MG63), lung cancer (Calu-6), colon cancer (HT-29), prostate cancer (PC3), and breast cancer (MDA-MB-231) (Figure 3). Interestingly, we found that concentrations of 1–10% of the original CM were very effective in inhibiting tumor cell growth and this effect was not augmented by increasing the concentration of CM added to the cultures. In addition, CM from foreskin fibroblasts failed to exhibit anti-tumor effects at any of the concentrations used (Figure 3B). To assess the anti-tumor efficacy of CM derived from BM-MSCs in vivo, we mixed CaLu-6 lung cancer cells with 10% CM and then injected the cells into immunocompromised mice. Remarkably, all 9 mice showed virtually no tumor formation when the CaLu-6 cells were mixed with CM prior to injection, while the mice injected with CaLu-6 cells mixed with PBS consistently developed tumors of various sizes (Figure 4).

In the present study, we also co-cultured tumor cells mixed with BM-MSCs at different ratios and injected tumor cells mixed with BM-MSCs into immunodeficient mice and observed that tumor cell growth was promoted in both in vitro and in vivo experiments when compared to tumor cells alone (data not shown). These results agree with those previously reported by others [19,20,21]. In addition, we found that MSCs harvested from low density (~50% confluent) versus high density (~100% confluent) cultures were able to stimulate cell growth to similar extents when mixed together with tumor cells and then co-cultured (data not shown). Our data supports the notion that MSC fate is controlled by the local microenvironment. As previously reported [22,23], MSCs, when mixed with tumor cells in a tumor-related microenvironment, differentiate into cancer-associated fibroblasts (CAFs) that produce VEGF, IL-6, TGFβ1, etc., resulting in pro-tumor activity. It is also shown that MSCs can release anti-tumor factors, including IGFBP, Dkk-1, miR-16, TRAIL, IFNα, and oncostatin-M [14,24]. In our study, tumor cell proliferation was almost completely inhibited when co-cultured directly or indirectly with confluent MSCs; moreover, CM derived from confluent BM-MSCs was also able to suppress tumor cell growth and increase tumor cell apoptosis at concentrations less than 30%. In addition, the anti-tumor activity of CM produced by confluent MSCs was further confirmed by our in vivo experiments. However, the inhibitory effect of sub-confluent MSCs in culture was not observed in our present animal studies. Taken together, these results suggest that confluent BM-MSCs predominantly exert anti-tumorigenic activity, which might be mediated by the secretion of DKK-1 and/or TRAIL, previoulsy shown to inhibit Wnt signaling, block tumor growth, and induce apoptosis in a variety of tumor cells [12,13,14]. The exact mechanisms, however, need to be investigated.

The results of the current study have led to a number of questions that need further research. For example, is it possible that the low amount of anti-tumor activity produced by sub-confluent (vs. confluent) MSCs in co-cultures and CM is due to reduced production or secretion of a different set of autocrine/paracrine factors? What is/are the key anti-tumor factor(s) produced by confluent MSCs? At present, it is unknown whether the anti-tumor factor(s) produced by confluent MSCs that inhibit tumor formation in vivo can also prevent tumor metastasis. To address these questions, we are in the process of performing proteomic studies to identify anti-tumor factors produced/secreted into the CM by MSCs and foreskin fibroblasts under confluent versus sub-confluent conditions. Subsequently, promising candidates (known or unknown) will be selected for further study, including the underlying molecular mechanisms and efficacy in treating various types of tumors in vivo using animal model systems.

In summary, our findings provide strong support for the hypothesis that contact inhibition in MSCs results in the production of autocrine/paracrine factors which also suppress tumor cell growth. These effectors display strong anti-tumor activity over a broad range of tumor types. More importantly, the use of these anti-tumor effectors as cell-free therapies in the future will address safety concerns caused by the use of live cells.

## Figures and Tables

**Figure 1 cells-12-02150-f001:**
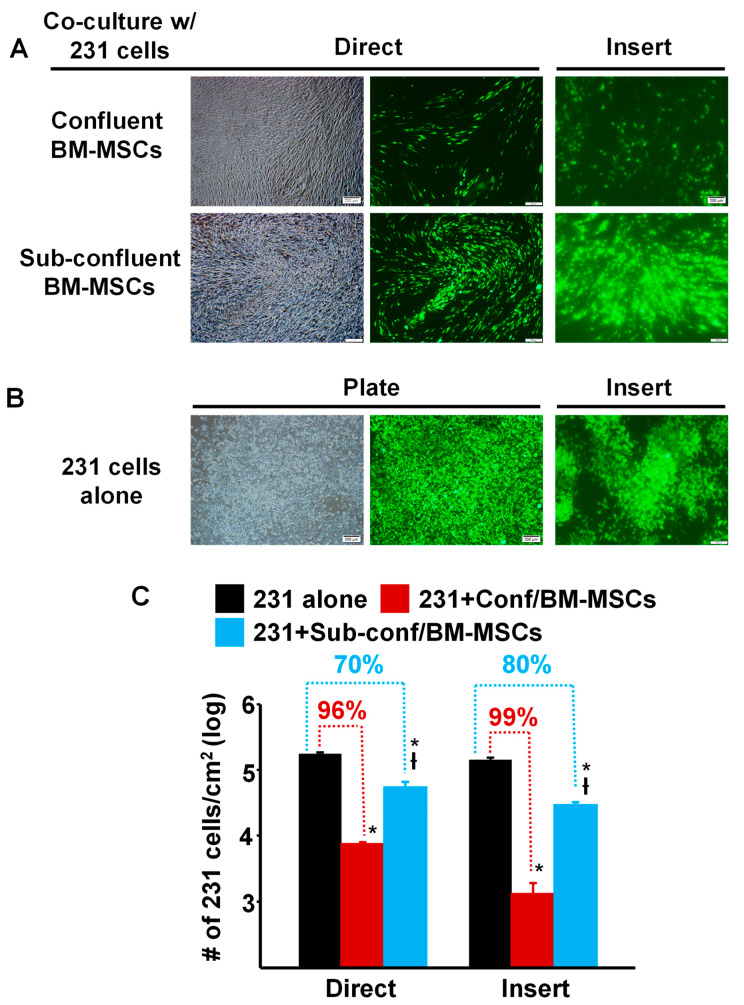
Proliferation of MDA-MB-231 cells co-cultured with confluent or sub-confluent BM-MSCs was dramatically decreased. (**A**) Phase contrast and fluorescence photomicrographs of MDA-MB-231 (231) cells (pre-labeled with green fluorescent protein) that were directly (Direct) co-cultured with passage 4 BM-MSCs or seeded onto a Transwell membrane and then placed into a well containing MSCs (“Insert”) and cultured for 5 days. **Upper panel:** MDA-MB-231 cells co-cultured with confluent BM-MSCs; and **Lower panel:** MDA-MB-231 cells co-cultured with sub-confluent BM-MSCs. Bar: 200 µm. Note: Focus of the cells in the fluorescence photomicrographs was difficult to obtain due to culture on the Transwell membrane. (**B**) Photomicrographs of MDA-MB-231 cells alone, not co-cultured, on the well of a tissue culture plate well or on a Transwell membrane (Insert). Bar: 200 µm. Note: Focus of the cells in the fluorescence photomicrographs was difficult to obtain due to culture on the Transwell membrane. (**C**) Quantitation of MDA-MB-231 cell growth after 5 days of co-culture (either Direct or on an Insert) with confluent or sub-confluent BM-MSCs. * *p* < 0.01 (*n* = 3) vs. 231 cells alone; † *p* < 0.01 (*n* = 3) vs. co-culture with confluent BM-MSCs.

**Figure 2 cells-12-02150-f002:**
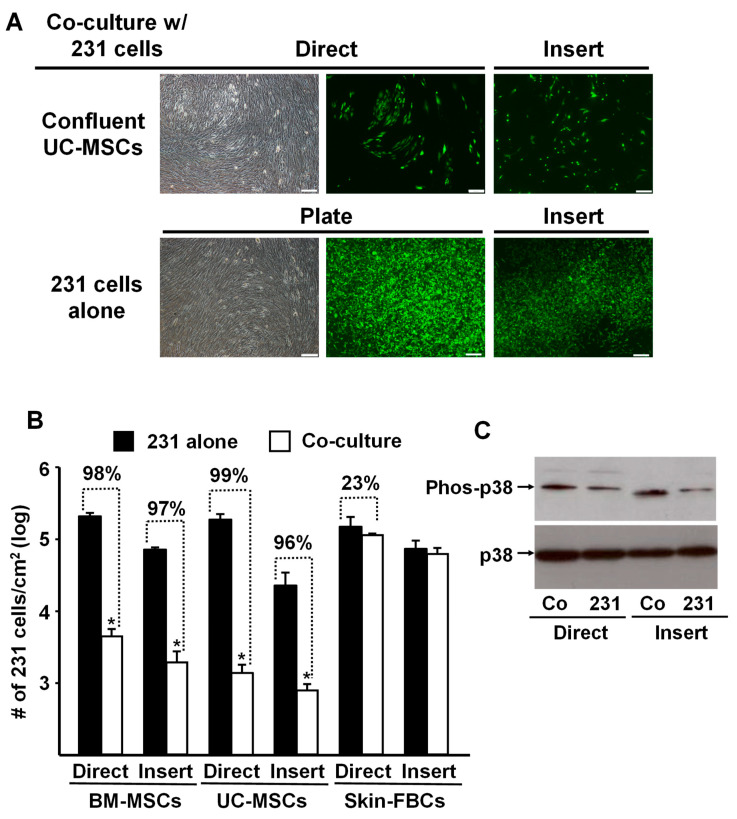
Proliferation of MDA-MB-231 cells was inhibited by co-culture with confluent BM- or UC- MSCs but not human neonatal foreskin fibroblasts. (**A**) Phase contrast and fluorescence photomicrographs of MDA-MB-231 cells (prelabeled with green fluorescent protein) that were directly (Direct) co-cultured with confluent UC-MSCs (passage 4) or seeded onto a Transwell membrane and then placed into a well containing UC-MSCs (“Insert”) and cultured for 5 days. Upper panel: MDA-MB-231 cells co-cultured with confluent UC-MSCs. Lower panel: MDA-MB-231 cells alone, not co-cultured, on the well of a tissue culture plate well or on a Transwell membrane (Insert). Bar: 200 µm. Note: Focus of the cells in the fluorescence photomicrographs was difficult to obtain due to culture on the Transwell membrane. (**B**) Quantitation of MDA-MB-231 cell growth after 5 days of “Direct” or “Insert” co-culture with confluent BM-MSCs, UC-MSCs, or Skin-FBCs. * *p* < 0.01 (*n* = 3) vs. MDA-MB-231 cells alone. (**C**) Western blot analysis of phospho-p38 up-regulation by MDA-MB-231 cells co-cultured with confluent BM-MSCs (Direct or on Inserts) for 5 days. Co: MDA-MB-231 cells co-cultured with BM-MSCs and 231: MDA-MB-231 cells cultured on TCP or Insert alone served as control. All experiments were repeated at least 3 times with MSCs from different donors, and the same results were obtained.

**Figure 3 cells-12-02150-f003:**
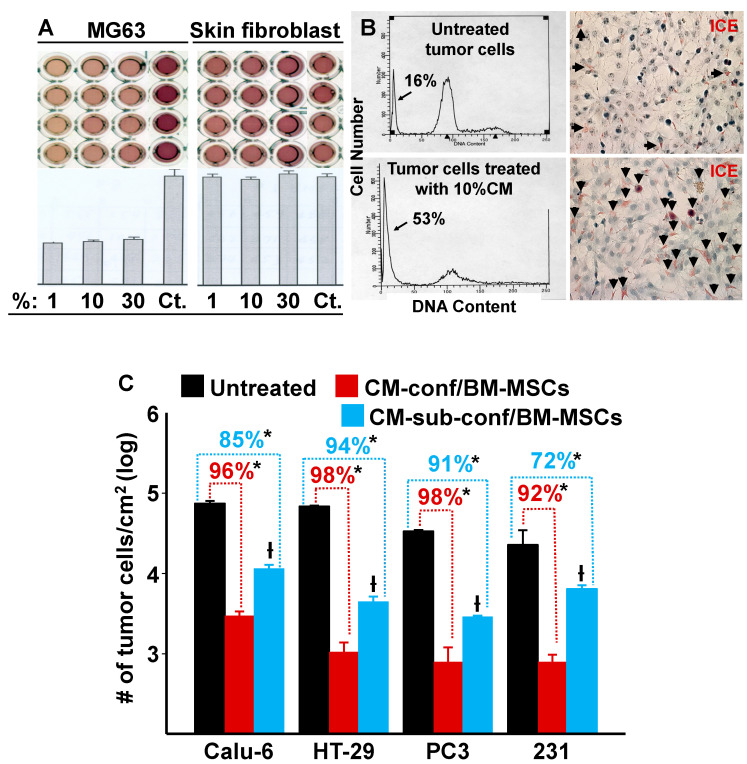
Conditioned media (CM) from cultures of BM-MSCs suppress the proliferation of various types of tumor cells but not human neonatal foreskin fibroblasts. (**A**) CM from cultures of confluent BM-MSCs were added at varying dilutions (1, 10, and 30%) to cultures of MG63 cells and neonatal foreskin fibroblasts. After incubation, MTT optical density (O.D.) was measured spectrophotometrically; a decrease in O.D. was proportional to an inhibition of cell growth/number. Controls (Ct) consisted of regular growth media that was used to dilute the CM. There was a marked inhibition of MG63 cell proliferation with the addition of CM; in contrast, no inhibition was observed with the foreskin fibroblasts. (**B**) Left panel: FACS analysis shows DNA content (arrow identifies the apoptotic population). Right panel: dead cells (stained red) were identified using the IL-1beta-converting enzyme (ICE) assay (arrows identify apoptotic cells). (**C**) The effect of 10% CM from confluent or sub-confluent BM-MSCs on the growth (cells/cm^2^) of a variety of tumor cell lines after 6 days in culture. * *p* < 0.01 (*n* = 3) vs. untreated cultures; † *p* < 0.01 (*n* = 3) vs. cells treated with CM from confluent cultures of BM-MSCs.

**Figure 4 cells-12-02150-f004:**
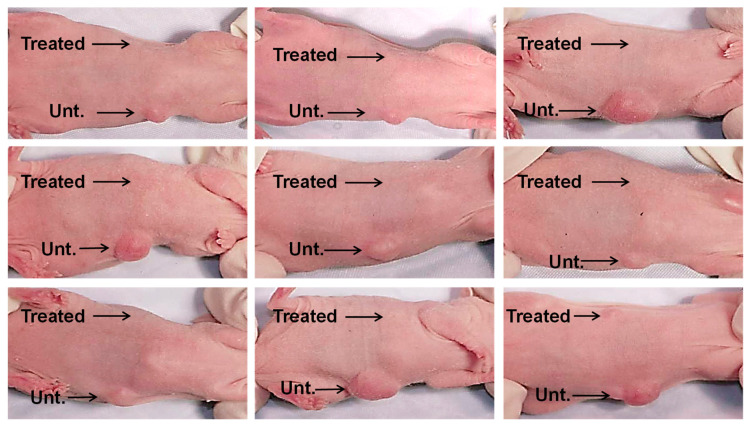
CM from BM-MSC cultures inhibit tumor formation in vivo. CaLu-6 cells were either left untreated (Unt.) with suspension in phosphate buffered saline (PBS) or treated (Treated) with suspension in 10% CM from confluent BM-MSC cultures and then injected subcutaneously into the ventral surface of 10-week-old immunodeficient beige mice at 1 × 10^6^ cells/injection. After 6 weeks, the mice were euthanized and photographed.

## Data Availability

The data presented in this study are available on request from the corresponding author.

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
