# Peer review of "Autocrine Factors Produced by Mesenchymal Stem Cells in Response to CellCell Contact Inhibition Have Anti-Tumor Properties"

_cells, 2023, doi:10.3390/cells12172150_

Round 1

Reviewer 1 Report

This interesting study  provides evidence that confluent MSCs produce inhibitory factor(s) that strongly suppress the proliferation of a variety of tumor cell lines in vitro as well as tumor growth in vivo. However is important to consider the following:

 1.- Provide more information, in the introduction and discussion section, on factors produced by MSCs that promote the inhibition of tumor cell proliferation, as well as the mechanisms involved in this phenomenon, which have been described in the literature.

2.- In the text,  lines 190-194, the authors describe that  -Interestingly, MDA-MB-231 cells co-cultured with confluent BM-MSCs up-regulated phosphorylated p38 (Phos-p38) MAPK (Fig. 2C), suggesting that the observed reduction in cancer cell proliferation might be associated with the up-regulated p38 phosphorylation and contact inhibition [13]- . However,  in the image of  Western blot analysis (Fig. 2C),     Phos-p38 of the  MDA-MB-231 cells co-cultured with confluent BM-MSCs (Direct or on Inserts) for 5 days appear to be diminished in comparison with those of  MDA-MB-231 cells cultured on TCP or  in Insert alone used  as control (Ct.),  which could indicate an opposite effect to that described by the authors, that is, MDA-MB-231 cells co-cultured with confluent BM-MSCs (direct or on Inserts) down-regulates Phos-p38. Please, clarify this point by including an inducer of p38 phosphorylation, as a positive control, in the culture of MDA-MB-231 cells.

Author Response

This interesting study provides evidence that confluent MSCs produce inhibitory factor(s) that strongly suppress the proliferation of a variety of tumor cell lines in vitro as well as tumor growth in vivo. However, is important to consider the following:

1.- Provide more information, in the introduction and discussion section, on factors produced by MSCs that promote the inhibition of tumor cell proliferation, as well as the mechanisms involved in this phenomenon, which have been described in the literature.

The reviewer’s suggestion has been well taken.  We have added more details about factors produced by MSCs to influence tumorigenesis on page 2 lines 53-62 in the Introduction and on page 9 lines 322-326 and lines 332-336 in the Discussion. 

2.- In the text, lines 190-194, the authors describe that Interestingly, MDA-MB-231 cells co-cultured with confluent BM-MSCs up-regulated phosphorylated p38 (Phos-p38) MAPK (Fig. 2C), suggesting that the observed reduction in cancer cell proliferation might be associated with the up-regulated p38 phosphorylation and contact inhibition [13]. However,  in the image of  Western blot analysis (Fig. 2C),     Phos-p38 of the  MDA-MB-231 cells co-cultured with confluent BM-MSCs (Direct or on Inserts) for 5 days appear to be diminished in comparison with those of  MDA-MB-231 cells cultured on TCP or  in Insert alone used  as control (Ct.),  which could indicate an opposite effect to that described by the authors, that is, MDA-MB-231 cells co-cultured with confluent BM-MSCs (direct or on Inserts) down-regulates Phos-p38. Please, clarify this point by including an inducer of p38 phosphorylation, as a positive control, in the culture of MDA-MB-231 cells.

We sincerely apologize for the mistakes.  “Ct” should have been labeled to indicate “Co-culture” and “231” to indicate “231 alone” (i.e. as a control).  This has been corrected in the revised manuscript, both in the figure and figure legend. 

Reviewer 2 Report

The authors have performed an interesting study that describes the importance of secreted components from confluent mesenchymal stem cells in regulating tumor cell growth. The study is quite promising and the use of secreted components of mesenchymal stem cells to treat cancer is of importance. However, a few additional experiments are needed to ascertain its importance.

Major comments

a) It would be interesting to get some clues on what is present in the secretomes of mesenchymal stem cells that could be causing the growth inhibition of cancer cells. To assess that authors should perform a secretome analysis of semi-confluent and fully confluent mesenchymal stem cells.

b) The in-vivo experiment is quite interesting. But in order to assess the therapeutic viability of this approach the best strategy will be to treat palpable tumors with mesenchymal stem cell conditioned media either by IP or IV or onsite injections and then assess tumor growth. This would greatly improve the significance of the paper.

Minor comments

 Figure 2C labeling is confusing. Looks like P38 phosphorylation is downregulated in treated groups and high in control groups. The opposite is described in the text. And if it is downregulated please test the effect of P38 inhibition on cell growth. 

The quality of English is good.

Author Response

The authors have performed an interesting study that describes the importance of secreted components from confluent mesenchymal stem cells in regulating tumor cell growth. The study is quite promising and the use of secreted components of mesenchymal stem cells to treat cancer is of importance. However, a few additional experiments are needed to ascertain its importance.

Major comments

  1. a) It would be interesting to get some clues on what is present in the secretomes of mesenchymal stem cells that could be causing the growth inhibition of cancer cells. To assess that authors should perform a secretome analysis of semi-confluent and fully confluent mesenchymal stem cells.

We really appreciate the reviewer’s comment and agree that it is very important experiment.  Proteomic studies are listed on page 10 lines 343-348 as one of our ongoing studies.  Because it is time consuming, we will include these results with other more comprehensive in vivo studies as the reviewer suggests below.   

  1. b) The in-vivo experiment is quite interesting. But in order to assess the therapeutic viability of this approach the best strategy will be to treat palpable tumors with mesenchymal stem cell conditioned media either by IP or IV or onsite injections and then assess tumor growth. This would greatly improve the significance of the paper.

Yes, we agree.  We did what the reviewer suggests.  The results were moderate, which could be caused by over diluting the inhibitory factor(s) in the conditioned media with a route of IP or IV injection.  We will repeat the in vivo studies with the purified candidates that may be selected as key effective factor(s) based on the proteomic analysis. 

 Minor comments

 Figure 2C labeling is confusing. Looks like P38 phosphorylation is downregulated in treated groups and high in control groups. The opposite is described in the text. And if it is down-regulated please test the effect of P38 inhibition on cell growth. 

We sincerely apologize for the mistakes.  “Ct” should have been labeled to indicate “Co-culture” and “231” to indicate “231 alone” (i.e. as a control).  This has been corrected in the revised manuscript, both in the figure and figure legend. 

Round 2

Reviewer 1 Report

All comments made to the manuscript were satisfactorily addressed.